# Local Privacy and Minimax Bounds: Sharp Rates for Probability Estimation

**John C. Duchi**[1]     **Michael I. Jordan**[1,2]     **Martin J. Wainwright**[1,2]
[1]Department of Electrical Engineering and Computer Science     [2]Department of Statistics
University of California, Berkeley
{jduchi,jordan,wainwrig}@eecs.berkeley.edu

## Abstract

We provide a detailed study of the estimation of probability distributions—discrete and continuous—in a stringent setting in which data is kept private even from the statistician. We give sharp minimax rates of convergence for estimation in these locally private settings, exhibiting fundamental trade-offs between privacy and convergence rate, as well as providing tools to allow movement along the privacy-statistical efficiency continuum. One of the consequences of our results is that Warner's classical work on randomized response is an optimal way to perform survey sampling while maintaining privacy of the respondents.

## 1   Introduction

The original motivation for providing privacy in statistical problems, first discussed by Warner [23], was that "for reasons of modesty, fear of being thought bigoted, or merely a reluctance to confide secrets to strangers," respondents to surveys might prefer to be able to answer certain questions non-truthfully, or at least without the interviewer knowing their true response. With this motivation, Warner considered the problem of estimating the fractions of the population belonging to certain strata, which can be viewed as probability estimation within a multinomial model. In this paper, we revisit Warner's probability estimation problem, doing so within a theoretical framework that allows us to characterize optimal estimation under constraints on privacy. We also apply our theoretical tools to a further probability estimation problem—that of nonparametric density estimation.

In the large body of research on privacy and statistical inference [e.g., 23, 14, 10, 15], a major focus has been on the problem of reducing disclosure risk: the probability that a member of a dataset can be identified given released statistics of the dataset. The literature has stopped short, however, of providing a formal treatment of disclosure risk that would permit decision-theoretic tools to be used in characterizing trade-offs between the utility of achieving privacy and the utility associated with an inferential goal. Recently, a formal treatment of disclosure risk known as "differential privacy" has been proposed and studied in the cryptography, database and theoretical computer science literatures [11, 1]. Differential privacy has strong semantic privacy guarantees that make it a good candidate for declaring a statistical procedure or data collection mechanism private, and it has been the focus of a growing body of recent work [13, 16, 24, 21, 6, 18, 8, 5, 9].

In this paper, we bring together the formal treatment of disclosure risk provided by differential privacy with the tools of minimax decision theory to provide a theoretical treatment of probability estimation under privacy constraints. Just as in classical minimax theory, we are able to provide lower bounds on the convergence rates of any estimator, in our case under a restriction to estimators that guarantee privacy. We complement these results with matching upper bounds that are achievable using computationally efficient algorithms. We thus bring classical notions of privacy, as introduced by Warner [23], into contact with differential privacy and statistical decision theory, obtaining quantitative trade-offs between privacy and statistical efficiency.

## 1.1 Setting and contributions

Let us develop some basic formalism before describing our main results. We study procedures that receive private views $Z_1, \ldots, Z_n \in \mathcal{Z}$ of an original set of observations, $X_1, \ldots, X_n \in \mathcal{X}$, where $\mathcal{X}$ is the (known) sample space. In our setting, $Z_i$ is drawn conditional on $X_i$ via the *channel distribution* $Q_i(Z_i \mid X_i = x)$; typically we omit the dependence of $Q_i$ on $i$. We focus in this paper on the non-interactive setting (in information-theoretic terms, on memoryless channels), where $Q_i$ is chosen prior to seeing data; see Duchi et al. [9] for more discussion.

We assume each of these private views $Z_i$ is $\alpha$-differentially private for the original data $X_i$. To give a precise definition for this type of privacy, known as "local privacy," let $\sigma(\mathcal{Z})$ be the $\sigma$-field on $\mathcal{Z}$ over which the channel $Q$ is defined. Then $Q$ provides $\alpha$-*local differential privacy* if

$$\sup \left\{ \frac{Q(S \mid X_i = x)}{Q(S \mid X_i = x')} \mid S \in \sigma(\mathcal{Z}), \text{ and } x, x' \in \mathcal{X} \right\} \leq \exp(\alpha). \tag{1}$$

This formulation of local privacy was first proposed by Evfimievski et al. [13]. The likelihood ratio bound (1) is attractive for many reasons. It means that any individual providing data guarantees his or her own privacy—no further processing or mistakes by a collection agency can compromise one's data—and the individual has plausible deniability about taking a value $x$, since any outcome $z$ is nearly as likely to have come from some other initial value $x'$. The likelihood ratio also controls the error rate in tests for the presence of points $x$ in the data [24].

In the current paper, we study minimax convergence rates when the data provided satisfies the local privacy guarantee (1). Our two main results quantify the penalty that must be paid when local privacy at a level $\alpha$ is provided in multinomial estimation and density estimation problems. At a high level, our first result implies that for estimation of a $d$-dimensional multinomial probability mass function, the effective sample size of *any* statistical estimation procedure decreases from $n$ to $n\alpha^2/d$ whenever $\alpha$ is a sufficiently small constant. A consequence of our results is that Warner's randomized response procedure [23] enjoys optimal sample complexity; it is interesting to note that even with the recent focus on privacy and statistical inference, the optimal privacy-preserving strategy for problems such as survey collection has been known for almost 50 years.

Our second main result, on density estimation, exhibits an interesting departure from standard minimax estimation results. If the density being estimated has $\beta$ continuous derivatives, then classical results on density estimation [e.g., 26, 25, 22] show that the minimax integrated squared error scales (in the sample size $n$) as $n^{-2\beta/(2\beta+1)}$. In the locally private case, we show that there is a difference in the *polynomial* rate of convergence: we obtain a scaling of $(\alpha^2 n)^{-2\beta/(2\beta+2)}$. We give efficiently implementable algorithms that attain sharp upper bounds as companions to our lower bounds, which in some cases exhibit the necessity of non-trivial sampling strategies to guarantee privacy.

**Notation:** Given distributions $P$ and $Q$ defined on a space $\mathcal{X}$, each absolutely continuous with respect to a measure $\mu$ (with densities $p$ and $q$), the KL-divergence between $P$ and $Q$ is

$$D_{\mathrm{kl}}(P \| Q) := \int_{\mathcal{X}} dP \log \frac{dP}{dQ} = \int_{\mathcal{X}} p \log \frac{p}{q} d\mu.$$

Letting $\sigma(\mathcal{X})$ denote an appropriate $\sigma$-field on $\mathcal{X}$, the total variation distance between $P$ and $Q$ is

$$\|P - Q\|_{\mathrm{TV}} := \sup_{S \in \sigma(\mathcal{X})} |P(S) - Q(S)| = \frac{1}{2} \int_{\mathcal{X}} |p(x) - q(x)| \, d\mu(x).$$

Let $X$ be distributed according to $P$ and $Y \mid X$ be distributed according to $Q(\cdot \mid X)$, and let $M = \int Q(\cdot \mid x) dP(x)$ denote the marginal of $Y$. The mutual information between $X$ and $Y$ is

$$I(X; Y) := \mathbb{E}_P \left[ D_{\mathrm{kl}}(Q(\cdot \mid X) \| M(\cdot)) \right] = \int D_{\mathrm{kl}}(Q(\cdot \mid X = x) \| M(\cdot)) \, dP(x).$$

A random variable $Y$ has Laplace$(\alpha)$ distribution if its density $p_Y(y) = \frac{\alpha}{2} \exp(-\alpha|y|)$. We write $a_n \lesssim b_n$ to denote $a_n = \mathcal{O}(b_n)$ and $a_n \asymp b_n$ to denote $a_n = \mathcal{O}(b_n)$ and $b_n = \mathcal{O}(a_n)$. For a convex set $C \subset \mathbb{R}^d$, we let $\Pi_C$ denote the orthogonal projection operator onto $C$.

## 2  Background and Problem Formulation

In this section, we provide the necessary background on the minimax framework used throughout the paper, more details of which can be found in standard sources [e.g., 17, 25, 26, 22]. We also reference our work [9] paper on statistical inference under differential privacy constraints; we restate two theorems from the paper [9] to keep our presentation self-contained.

### 2.1  Minimax framework

Let $\mathcal{P}$ denote a class of distributions on the sample space $\mathcal{X}$, and let $\theta : \mathcal{P} \to \Theta$ denote a function defined on $\mathcal{P}$. The range $\Theta$ depends on the underlying statistical model; for example, for density estimation, $\Theta$ may consist of the set of probability densities defined on $[0, 1]$. We let $\rho$ denote the semi-metric on the space $\Theta$ that we use to measure the error of an estimator for $\theta$, and $\Phi : \mathbb{R}_+ \to \mathbb{R}_+$ be a non-decreasing function with $\Phi(0) = 0$ (for example, $\Phi(t) = t^2$).

Recalling that $\mathcal{Z}$ is the domain of the private variables $Z_i$, let $\widehat{\theta} : \mathcal{Z}^n \to \Theta$ denote an arbitrary estimator for $\theta$. Let $\mathcal{Q}_\alpha$ denote the set of conditional (or channel) distributions guaranteeing $\alpha$-local privacy (1). Looking uniformly over all channels $Q \in \mathcal{Q}_\alpha$, we define the central object of interest for this paper, the $\alpha$-*private minimax rate* for the family $\theta(\mathcal{P})$,

$$\mathfrak{M}_n(\theta(\mathcal{P}), \Phi \circ \rho, \alpha) := \inf_{\widehat{\theta}, Q \in \mathcal{Q}_\alpha} \sup_{P \in \mathcal{P}} \mathbb{E}_{P,Q} \left[ \Phi \left( \rho(\widehat{\theta}(Z_1, \ldots, Z_n), \theta(P)) \right) \right]. \tag{2}$$

associated with estimating $\theta$ based on $(Z_1, \ldots, Z_n)$. We remark here (see also the discussion in [9]) that the private minimax risk (2) is different from previous work on optimality in differential privacy (e.g. [2, 16, 8]): prior work focuses on accurate estimation of a *sample* quantity $\theta(x_{1:n})$ based on the sample $x_{1:n}$, while we provide lower bounds on error of the *population* estimator $\theta(P)$. Lower bounds on population estimation imply those on sample estimation, so our lower bounds are stronger than most of those in prior work.

A standard route for lower bounding the minimax risk (2) is by reducing the estimation problem to the testing problem of identifying a point $\theta \in \Theta$ from a collection of well-separated points [26, 25]. Given an index set $\mathcal{V}$, the indexed family of distributions $\{P_\nu, \nu \in \mathcal{V}\} \subset \mathcal{P}$ is a $2\delta$-packing of $\Theta$ if $\rho(\theta(P_\nu), \theta(P_{\nu'})) \geq 2\delta$ for all $\nu \neq \nu'$ in $\mathcal{V}$. The setup is that of a standard hypothesis testing problem: nature chooses $V \in \mathcal{V}$ uniformly at random, then data $(X_1, \ldots, X_n)$ are drawn i.i.d. from $P_\nu^n$, conditioning on $V = \nu$. The problem is to identify the member $\nu$ of the packing set $\mathcal{V}$.

In this work we have the additional complication that all the statistician observes are the private samples $Z_1, \ldots, Z_n$. To that end, if we let $Q^n(\cdot \mid x_{1:n})$ denote the conditional distribution of $Z_1, \ldots, Z_n$ given that $X_1 = x_1, \ldots, X_n = x_n$, we define the marginal channel $M_\nu^n$ via the expression

$$M_\nu^n(A) := \int Q^n(A \mid x_1, \ldots, x_n) dP_\nu(x_1, \ldots, x_n) \quad \text{for } A \in \sigma(\mathcal{Z}^n). \tag{3}$$

Letting $\psi : \mathcal{Z}^n \to \mathcal{V}$ denote an arbitrary testing procedure, we have the following minimax bound, whose two parts are known as Le Cam's two-point method [26, 22] and Fano's inequality [25, 7, 22].

**Lemma 1** (Minimax risk bound). *For the previously described estimation and testing problems,*

$$\mathfrak{M}_n(\theta(\mathcal{P}), \Phi \circ \rho, Q) \geq \Phi(\delta) \inf_\psi \mathbb{P}(\psi(Z_1, \ldots, Z_n) \neq V), \tag{4}$$

*where the infimum is taken over all testing procedures. For a binary test specified by $\mathcal{V} = \{\nu, \nu'\}$,*

$$\inf_\psi \mathbb{P}(\psi(Z_1, \ldots, Z_n) \neq V) = \frac{1}{2} - \frac{1}{2} \|M_\nu^n - M_{\nu'}^n\|_{\mathrm{TV}}, \tag{5a}$$

*and more generally,*

$$\inf_\psi \mathbb{P}(\psi(Z_1, \ldots, Z_n) \neq V) \geq \left[ 1 - \frac{I(Z_1, \ldots, Z_n; V) + \log 2}{\log |\mathcal{V}|} \right]. \tag{5b}$$

## 2.2 Information bounds

The main step in proving minimax lower bounds is to control the divergences involved in the lower bounds (5a) and (5b). We review two results from our work [9] that obtain such bounds as a function of the amount of privacy provided. The second of the results provides a variational upper bound on the mutual information $I(Z_1, \ldots, Z_n; V)$, in that we optimize jointly over subset $S \subset \mathcal{X}$. To state the proposition, we require a bit of notation: for each $i \in \{1, \ldots, n\}$, let $P_{\nu,i}$ be the distribution of $X_i$ conditional on the random packing element $V = \nu$, and let $M_\nu^n$ be the marginal distribution (3) induced by passing $X_i$ through $Q$. Define the mixture distribution $\overline{P}_i = \frac{1}{|\mathcal{V}|} \sum_{\nu \in \mathcal{V}} P_{\nu,i}$, We can then state a proposition summarizing the results we require from Duchi et al. [9]:

**Proposition 1** (Information bounds). *For any $\nu, \nu' \in \mathcal{V}$ and $\alpha \geq 0$,*

$$D_{\mathrm{kl}} \left( M_\nu^n \| M_{\nu'}^n \right) \leq 4(e^\alpha - 1)^2 \sum_{i=1}^n \| P_{\nu,i} - P_{\nu',i} \|_{\mathrm{TV}}^2 . \tag{6}$$

*Additionally for $V$ chosen uniformly at random from $\mathcal{V}$, we have the variational bound*

$$I(Z_1, \ldots, Z_n; V) \leq e^\alpha \frac{(e^\alpha - e^{-\alpha})^2}{|\mathcal{V}|} \sum_{i=1}^n \sup_{S \in \sigma(\mathcal{X})} \sum_{\nu \in \mathcal{V}} \left( P_{\nu,i}(S) - \overline{P}(S) \right)^2 . \tag{7}$$

By combining Proposition 1 with Lemma 1, it is possible to derive sharp lower bounds on arbitrary estimation procedures under $\alpha$-local privacy. In the remainder of the paper, we demonstrate this combination for probability estimation problems; we provide proofs of all results in [9].

# 3  Multinomial Estimation under Local Privacy

In this section we return to the classical problem of avoiding answer bias in surveys, the original motivation for studying local privacy [23].

## 3.1  Minimax rates of convergence for multinomial estimation

Let $\Delta_d := \left\{ \theta \in \mathbb{R}^d \mid \theta \geq 0, \sum_{j=1}^d \theta_j = 1 \right\}$ denote the probability simplex in $\mathbb{R}^d$. The multinomial estimation problem is defined as follows. Given a vector $\theta \in \Delta_d$, samples $X$ are drawn i.i.d. from a multinomial with parameters $\theta$, where $P_\theta(X = j) = \theta_j$ for $j \in \{1, \ldots, d\}$, and the goal is to estimate $\theta$. In one of the earliest evaluations of privacy, Warner [23] studied the Bernoulli variant of this problem and proposed *randomized response*: for a given survey question, respondents provide a truthful answer with probability $p > 1/2$ and lie with probability $1 - p$.

In our setting, we assume the statistician sees $\alpha$-locally private (1) random variables $Z_i$ for the corresponding samples $X_i$ from the multinomial. In this case, we have the following result, which characterizes the minimax rate of estimation of a multinomial in both mean-squared error $\mathbb{E}[\|\widehat{\theta} - \theta\|_2^2]$ and absolute error $\mathbb{E}[\|\widehat{\theta} - \theta\|_1]$; the latter may be more relevant for probability estimation problems.

**Theorem 1.** *There exist universal constants $0 < c_\ell \leq c_u < 5$ such that for all $\alpha \in [0, 1]$, the minimax rate for multinomial estimation satisfies the bounds*

$$c_\ell \min \left\{ 1, \frac{1}{\sqrt{n\alpha^2}}, \frac{d}{n\alpha^2} \right\} \leq \mathfrak{M}_n \left( \Delta_d, \|\cdot\|_2^2, \alpha \right) \leq c_u \min \left\{ 1, \frac{d}{n\alpha^2} \right\}, \tag{8}$$

*and*

$$c_\ell \min \left\{ 1, \frac{d}{\sqrt{n\alpha^2}} \right\} \leq \mathfrak{M}_n \left( \Delta_d, \|\cdot\|_1, \alpha \right) \leq c_u \min \left\{ 1, \frac{d}{\sqrt{n\alpha^2}} \right\}. \tag{9}$$

Theorem 1 shows that providing local privacy can sometimes be quite detrimental to the quality of statistical estimators. Indeed, let us compare this rate to the classical rate in which there is no privacy. Then estimating $\theta$ via proportions (i.e., maximum likelihood), we have

$$\mathbb{E} \left[ \|\widehat{\theta} - \theta\|_2^2 \right] = \sum_{j=1}^d \mathbb{E} \left[ (\widehat{\theta}_j - \theta_j)^2 \right] = \frac{1}{n} \sum_{j=1}^d \theta_j (1 - \theta_j) \leq \frac{1}{n} \left( 1 - \frac{1}{d} \right) < \frac{1}{n}.$$

By inequality (8), for suitably large sample sizes $n$, the effect of providing differential privacy at a level $\alpha$ causes a reduction in the effective sample size of $n \mapsto n\alpha^2/d$.

## 3.2 Optimal mechanisms: attainability for multinomial estimation

An interesting consequence of the lower bound in (8) is the following fact that we now demonstrate: Warner's classical randomized response mechanism [23] (with minor modification) achieves the optimal convergence rate. There are also other relatively simple estimation strategies that achieve convergence rate $d/n\alpha^2$; the perturbation approach Dwork et al. [11] propose, where $\mathrm{Laplace}(\alpha)$ noise is added to each coordinate of a multinomial sample, is one such strategy. Nonetheless, the ease of use and explainability of randomized response, coupled with our optimality results, provide support for randomized response as a preferred method for private estimation of population probabilities.

We now prove that randomized response attains the optimal rate of convergence. There is a bijection between multinomial samples $x \in \{1, \dots, d\}$ and the $d$ standard basis vectors $e_1, \dots, e_d \in \mathbb{R}^d$, so we abuse notation and represent samples $x$ as either when designing estimation strategies. In randomized response, we construct the private vector $Z \in \{0,1\}^d$ from a multinomial observation $x \in \{e_1, \dots, e_d\}$ by sampling $d$ coordinates independently via the procedure

$$[Z]_j = \begin{cases} x_j & \text{with probability } \frac{\exp(\alpha/2)}{1+\exp(\alpha/2)} \\ 1 - x_j & \text{with probability } \frac{1}{1+\exp(\alpha/2)}. \end{cases} \tag{10}$$

We claim that this channel (10) is $\alpha$-differentially private: indeed, note that for any $x, x' \in \Delta_d$ and any vector $z \in \{0,1\}^d$ we have

$$\frac{Q(Z = z \mid x)}{Q(Z = z \mid x')} = \exp\left(\frac{\alpha}{2}\left(\|z - x\|_1 - \|z - x'\|_1\right)\right) \in [\exp(-\alpha), \exp(\alpha)],$$

where we used the triangle inequality to assert that $|\|z - x\|_1 - \|z - x'\|_1| \le \|x - x'\|_1 \le 2$. We can compute the expected value and variance of the random variables $Z$; indeed, by definition (10)

$$\mathbb{E}[Z \mid x] = \frac{e^{\alpha/2}}{1 + e^{\alpha/2}} x + \frac{1}{1 + e^{\alpha/2}} (\mathbb{1} - x) = \frac{e^{\alpha/2} - 1}{e^{\alpha/2} + 1} x + \frac{1}{1 + e^{\alpha/2}} \mathbb{1}.$$

Since the $Z$ are Bernoulli, we obtain the variance bound $\mathbb{E}[\|Z - \mathbb{E}[Z]\|_2^2] < d/4 + 1 < d$. Recalling the definition of the projection $\Pi_{\Delta_d}$ onto the simplex, we arrive at the natural estimator

$$\widehat{\theta}_{\text{part}} := \frac{1}{n} \sum_{i=1}^{n} \left(Z_i - \mathbb{1}/(1 + e^{\alpha/2})\right) \frac{e^{\alpha/2} + 1}{e^{\alpha/2} - 1} \quad \text{and} \quad \widehat{\theta} := \Pi_{\Delta_d}\left(\widehat{\theta}_{\text{part}}\right). \tag{11}$$

The projection of $\widehat{\theta}_{\text{part}}$ onto the probability simplex can be done in time linear in the dimension $d$ of the problem [3], so the estimator (11) is efficiently computable. Since projections only decrease distance, vectors in the simplex are at most distance $\sqrt{2}$ apart, and $\mathbb{E}_\theta[\widehat{\theta}_{\text{part}}] = \theta$, we find

$$\mathbb{E}\left[\|\widehat{\theta} - \theta\|_2^2\right] \le \min\left\{2, \mathbb{E}\left[\|\widehat{\theta}_{\text{part}} - \theta\|_2^2\right]\right\} \le \min\left\{2, \frac{d}{n}\left(\frac{e^{\alpha/2} + 1}{e^{\alpha/2} - 1}\right)^2\right\} \lesssim \min\left\{1, \frac{d}{n\alpha^2}\right\}.$$

A similar argument shows that randomized response is minimax optimal for the $\ell_1$-loss as well.

## 4 Density Estimation under Local Privacy

In this section, we turn to studying a nonparametric statistical problem in which the effects of local differential privacy turn out to be somewhat more severe. We show that for the problem of density estimation, instead of just multiplicative loss in the effective sample size as in the previous section, imposing local differential privacy leads to a different convergence rate.

In more detail, we consider estimation of probability densities $f : \mathbb{R} \to \mathbb{R}_+$, $\int f(x) dx = 1$ and $f \ge 0$, defined on the real line, focusing on a standard family of densities of varying smoothness [e.g. 22]. Throughout this section, we let $\beta \in \mathbb{N}$ denote a fixed positive integer. Roughly, we consider densities that have bounded $\beta$th derivative, and we study density estimation using the squared $L^2$-norm $\|f\|_2^2 := \int f^2(x) dx$ as our metric; in formal terms, we impose these constraints in terms of Sobolev classes (e.g. [22, 12]). Let the countable collection of functions $\{\varphi_j\}_{j=1}^{\infty}$ be an orthonormal basis for $L^2([0,1])$. Then any function $f \in L^2([0,1])$ can be expanded as a sum $\sum_{j=1}^{\infty} \theta_j \varphi_j$ in terms of the basis coefficients $\theta_j := \int f(x) \varphi_j(x) dx$, where $\{\theta_j\}_{j=1}^{\infty} \in \ell^2(\mathbb{N})$. The *Sobolev space* $\mathcal{F}_\beta[C]$ is obtained by enforcing a particular decay rate on the coefficients $\theta$:

**Definition 1** (Elliptical Sobolev space). For a given orthonormal basis $\{\varphi_j\}$ of $L^2([0,1])$, smoothness parameter $\beta > 1/2$ and radius $C$, the function class $\mathcal{F}_\beta[C]$ is given by

$$\mathcal{F}_\beta[C] := \left\{ f \in L^2([0,1]) \mid f = \sum_{j=1}^\infty \theta_j \varphi_j \text{ such that } \sum_{j=1}^\infty j^{2\beta} \varphi_j^2 \le C^2 \right\}.$$

If we choose the trigonometric basis as our orthonormal basis, then membership in the class $\mathcal{F}_\beta[C]$ corresponds to certain smoothness constraints on the derivatives of $f$. More precisely, for $j \in \mathbb{N}$, consider the orthonormal basis for $L^2([0,1])$ of trigonometric functions:

$$\varphi_0(t) = 1, \quad \varphi_{2j}(t) = \sqrt{2}\cos(2\pi j t), \quad \varphi_{2j+1}(t) = \sqrt{2}\sin(2\pi j t). \tag{12}$$

Now consider a $\beta$-times almost everywhere differentiable function $f$ for which $|f^{(\beta)}(x)| \le C$ for almost every $x \in [0,1]$ satisfying $f^{(k)}(0) = f^{(k)}(1)$ for $k \le \beta - 1$. Uniformly for such $f$, there is a universal constant $c$ such that that $f \in \mathcal{F}_\beta[cC]$ [22, Lemma A.3]. Thus, Definition 1 (essentially) captures densities that have Lipschitz-continuous $(\beta - 1)$th derivative. In the sequel, we write $\mathcal{F}_\beta$ when the bound $C$ in $\mathcal{F}_\beta[C]$ is $\mathcal{O}(1)$. It is well known [26, 25, 22] that the minimax risk for non-private estimation of densities in the class $\mathcal{F}_\beta$ scales as

$$\mathfrak{M}_n \left( \mathcal{F}_\beta, \|\cdot\|_2^2, \infty \right) \asymp n^{-\frac{2\beta}{2\beta+1}}. \tag{13}$$

Our main result is to demonstrate that the classical rate (13) is *no longer attainable* when we require $\alpha$-local differential privacy. In Sections 4.2 and 4.3, we show how to achieve the (new) optimal rate using histogram and orthogonal series estimators.

## 4.1 Lower bounds on density estimation

We begin by giving our main lower bound on the minimax rate of estimation of densities when are kept differentially private, providing the proof in the longer paper [9].

**Theorem 2.** *Consider the class of densities $\mathcal{F}_\beta$ defined using the trigonometric basis (12). For some $\alpha \in [0,1]$, suppose $Z_i$ are $\alpha$-locally private (1) for the samples $X_i \in [0,1]$. There exists a constant $c_\beta > 0$, dependent only on $\beta$, such that*

$$\mathfrak{M}_n \left( \mathcal{F}_\beta, \|\cdot\|_2^2, \alpha \right) \ge c_\beta \left( n\alpha^2 \right)^{-\frac{2\beta}{2\beta+2}}. \tag{14}$$

In comparison with the classical minimax rate (13), the lower bound (14) involves a different polynomial exponent: privacy reduces the exponent from $2\beta/(2\beta+1)$ to $2\beta/(2\beta+2)$. For example, for Lipschitz densities we have $\beta = 1$, and the rate degrades from $n^{-2/3}$ to $n^{-1/2}$.

Interestingly, no estimator based on Laplace (or exponential) perturbation of the samples $X_i$ themselves can attain the rate of convergence (14). In their study of the deconvolution problem, Carroll and Hall [4] show that if samples $X_i$ are perturbed by additive noise $W$, where the characteristic function $\phi_W$ of the additive noise has tails behaving as $|\phi_W(t)| = \mathcal{O}(|t|^{-a})$ for some $a > 0$, then no estimator can deconvolve the samples $X + W$ and attain a rate of convergence better than $n^{-2\beta/(2\beta+2a+1)}$. Since the Laplace distribution's characteristic function has tails decaying as $t^{-2}$, no estimator based on perturbing the samples directly can attain a rate of convergence better than $n^{-2\beta/(2\beta+5)}$. If the lower bound (14) is attainable, we must then study privacy mechanisms that are not simply based on direct perturbation of the samples $\{X_i\}_{i=1}^n$.

## 4.2 Achievability by histogram estimators

We now turn to the mean-squared errors achieved by specific practical schemes, beginning with the special case of Lipschitz density functions ($\beta = 1$), for which it suffices to consider a private version of a classical histogram estimate. For a fixed positive integer $k \in \mathbb{N}$, let $\{\mathcal{X}_j\}_{j=1}^k$ denote the partition of $\mathcal{X} = [0,1]$ into the intervals

$$\mathcal{X}_j = [(j-1)/k, j/k) \quad \text{for } j = 1, 2, \dots, k-1, \text{ and } \mathcal{X}_k = [(k-1)/k, 1].$$

Any histogram estimate of the density based on these $k$ bins can be specified by a vector $\theta \in k\Delta_k$, where we recall $\Delta_k \subset \mathbb{R}_+^k$ is the probability simplex. Any such vector defines a density estimate via the sum $f_\theta := \sum_{j=1}^k \theta_j 1_{\mathcal{X}_j}$, where $1_E$ denotes the characteristic (indicator) function of the set $E$.

Let us now describe a mechanism that guarantees $\alpha$-local differential privacy. Given a data set $\{X_1, \ldots, X_n\}$ of samples from the distribution $f$, consider the vectors

$$Z_i := \mathsf{e}_k(X_i) + W_i, \quad \text{for } i = 1, 2, \ldots, n, \tag{15}$$

where $\mathsf{e}_k(X_i) \in \Delta_k$ is a $k$-vector with the $j$th entry equal to one if $X_i \in \mathcal{X}_j$, and zeroes in all other entries, and $W_i$ is a random vector with i.i.d. $\mathrm{Laplace}(\alpha/2)$ entries. The variables $\{Z_i\}_{i=1}^n$ so-defined are $\alpha$-locally differentially private for $\{X_i\}_{i=1}^n$.

Using these private variables, we then form the density estimate $\widehat{f} := f_{\widehat{\theta}} = \sum_{j=1}^k \widehat{\theta}_j 1_{\mathcal{X}_j}$ based on

$$\widehat{\theta} := \Pi_k \left( \frac{k}{n} \sum_{i=1}^n Z_i \right), \tag{16}$$

where $\Pi_k$ denotes the Euclidean projection operator onto the set $k\Delta_k$. By construction, we have $\widehat{f} \geq 0$ and $\int_0^1 \widehat{f}(x)dx = 1$, so $\widehat{f}$ is a valid density estimate.

**Proposition 2.** *Consider the estimate $\widehat{f}$ based on $k = (n\alpha^2)^{1/4}$ bins in the histogram. For any 1-Lipschitz density $f : [0,1] \to \mathbb{R}_+$, we have*

$$\mathbb{E}_f \left[ \left\| \widehat{f} - f \right\|_2^2 \right] \leq 5(\alpha^2 n)^{-\frac{1}{2}} + \sqrt{\alpha} n^{-3/4}. \tag{17}$$

For any fixed $\alpha > 0$, the first term in the bound (17) dominates, and the $\mathcal{O}((\alpha^2 n)^{-\frac{1}{2}})$ rate matches the minimax lower bound (14) in the case $\beta = 1$: the privatized histogram estimator is minimax-optimal for Lipschitz densities. This result provides the private analog of the classical result that histogram estimators are minimax-optimal (in the non-private setting) for Lipschitz densities.

## 4.3 Achievability by orthogonal projection estimators

For higher degrees of smoothness ($\beta > 1$), histogram estimators no longer achieve optimal rates in the classical setting [20]. Accordingly, we turn to estimators based on orthogonal series and show that even under local privacy, they achieve the lower bound (14) for all orders of smoothness $\beta \geq 1$.

Recall the elliptical Sobolev space (Definition 1), in which a function $f$ is represented as $f = \sum_{j=1}^\infty \theta_j \varphi_j$, where $\theta_j = \int f(x)\varphi_j(x)dx$. This representation underlies the classical method of orthonormal series estimation: given a data set, $\{X_1, X_2, \ldots, X_n\}$, drawn i.i.d. according to a density $f \in L^2([0,1])$, we first compute the empirical basis coefficients

$$\widehat{\theta}_j = \frac{1}{n} \sum_{i=1}^n \varphi_j(X_i) \quad \text{and then set} \quad \widehat{f} = \sum_{j=1}^k \widehat{\theta}_j \varphi_j, \tag{18}$$

where the value $k \in \mathbb{N}$ is chosen either a priori based on known properties of the estimation problem or adaptively, for example, using cross-validation [12, 22].

In the setting of local privacy, we consider a mechanism that, instead of releasing the vector of coefficients $(\varphi_1(X_i), \ldots, \varphi_k(X_i))$ for each data point, employs a random vector $Z_i = (Z_{i,1}, \ldots, Z_{i,k})$ with the property that $\mathbb{E}[Z_{i,j} \mid X_i] = \varphi_j(X_i)$ for each $j = 1, 2, \ldots, k$. We assume the basis functions are uniformly bounded; i.e., there exists a constant $B_0 = \sup_j \sup_x |\varphi_j(x)| < \infty$. For a fixed number $B$ strictly larger than $B_0$ (to be specified momentarily), consider the following scheme:

**Sampling strategy** Given a vector $\tau \in [-B_0, B_0]^k$, construct $\widetilde{\tau} \in \{-B_0, B_0\}^k$ with coordinates $\widetilde{\tau}_j$ sampled independently from $\{-B_0, B_0\}$ with probabilities $\frac{1}{2} - \frac{\tau_j}{2B_0}$ and $\frac{1}{2} + \frac{\tau_j}{2B_0}$. Sample $T$ from a $\mathrm{Bernoulli}(e^\alpha/(e^\alpha + 1))$ distribution. Then choose $Z \in \{-B, B\}^k$ via

$$Z \sim \begin{cases} \text{Uniform on } \left\{ z \in \{-B, B\}^k : \langle z, \widetilde{\tau} \rangle > 0 \right\} & \text{if } T = 1 \\ \text{Uniform on } \left\{ z \in \{-B, B\}^k : \langle z, \widetilde{\tau} \rangle \leq 0 \right\} & \text{if } T = 0. \end{cases} \tag{19}$$

By inspection, $Z$ is $\alpha$-differentially private for any initial vector in the box $[-B_0, B_0]^k$, and moreover, the samples (19) are efficiently computable (for example by rejection sampling). Starting from the vector $\tau \in \mathbb{R}^k$, $\tau_j = \varphi_j(X_i)$, in the above sampling strategy we have

$$\mathbb{E}[[Z]_j \mid X = x] = c_k \frac{B}{B_0\sqrt{k}} \left( \frac{e^\alpha}{e^\alpha + 1} - \frac{1}{e^\alpha + 1} \right) \varphi_j(x) = c_k \frac{B}{B_0\sqrt{k}} \frac{e^\alpha - 1}{e^\alpha + 1} \varphi_j(x), \qquad (20)$$

for a constant $c_k$ that may depend on $k$ but is $\mathcal{O}(1)$ and bounded away from 0. Consequently, to attain the unbiasedness condition $\mathbb{E}[[Z_i]_j \mid X_i] = \varphi_j(X_i)$, it suffices to take $B = \mathcal{O}(B_0\sqrt{k}/\alpha)$.

The full sampling and inferential scheme are as follows: (i) given a data point $X_i$, construct the vector $\tau = [\varphi_j(X_i)]_{j=1}^k$; (ii) sample $Z_i$ according to strategy (19) using $\tau$ and the bound $B = B_0\sqrt{k}(e^\alpha + 1)/c_k(e^\alpha - 1)$. (The constant $c_k$ is as in the expression (20).) Using the estimator

$$\widehat{f} := \frac{1}{n} \sum_{i=1}^n \sum_{j=1}^k Z_{i,j} \varphi_j, \qquad (21)$$

we obtain the following proposition.

**Proposition 3.** *Let $\{\varphi_j\}$ be a $B_0$-bounded orthonormal basis for $L^2([0,1])$. There exists a constant $c$ (depending only on $C$ and $B_0$) such that the estimator* (21) *with $k = (n\alpha^2)^{1/(2\beta+2)}$ satisfies*

$$\sup_{f \in \mathcal{F}_\beta[C]} \mathbb{E}_f \left[ \|f - \widehat{f}\|_2^2 \right] \leq c \left( n\alpha^2 \right)^{-\frac{2\beta}{2\beta+2}}.$$

Propositions 2 and 3 make clear that the minimax lower bound (14) is sharp, as claimed.

Before concluding our exposition, we make a few remarks on other potential density estimators. Our orthogonal-series estimator (21) (and sampling scheme (20)), while similar in spirit to that proposed by Wasserman and Zhou [24, Sec. 6], is different in that it is locally private and requires a different noise strategy to obtain both $\alpha$-local privacy and optimal convergence rate. Lei [19] considers private $M$-estimators based on first performing a histogram density estimate, then using this to construct a second estimator; his estimator is not locally private, and the resulting $M$-estimators have sub-optimal convergence rates. Finally, we remark that density estimators that are based on orthogonal series and Laplace perturbation are sub-optimal: they can achieve (at best) rates of $(n\alpha^2)^{-\frac{2\beta}{2\beta+3}}$, which is polynomially worse than the sharp result provided by Proposition 3. It appears that appropriately chosen noise mechanisms are crucial for obtaining optimal results.

## 5 Discussion

We have linked minimax analysis from statistical decision theory with differential privacy, bringing some of their respective foundational principles into close contact. In this paper particularly, we showed how to apply our divergence bounds to obtain sharp bounds on the convergence rate for certain nonparametric problems in addition to standard finite-dimensional settings. By providing sharp convergence rates for many standard statistical inference procedures under local differential privacy, we have developed and explored some tools that may be used to better understand privacy-preserving statistical inference and estimation procedures. We have identified a fundamental continuum along which privacy may be traded for utility in the form of accurate statistical estimates, providing a way to adjust statistical procedures to meet the privacy or utility needs of the statistician and the population being sampled. Formally identifying this trade-off in other statistical problems should allow us to better understand the costs and benefits of privacy; we believe we have laid some of the groundwork to do so.

**Acknowledgments**

JCD was supported by a Facebook Graduate Fellowship and an NDSEG fellowship. Our work was supported in part by the U.S. Army Research Laboratory, U.S. Army Research Office under grant number W911NF-11-1-0391, and Office of Naval Research MURI grant N00014-11-1-0688.

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
