[Reviews · NeurIPS 2013]

Submitted by Assigned_Reviewer_7

SUMMARY: This paper is a NIPS-formatted version of an ArXiV manuscript, and uses a Fano/LeCam-style argument to derive a lower bound on estimation algorithms that operate on private data when the algorithm is not trusted by the data holder. As a corollary, randomized response turns out to be an optimal strategy in some sense.

As a caveat to this review, I did not go through the supplementary material.

pros and cons:
- the results provide characterize the limitations of learning from data that has been perturbed to guarantee privacy
- there is some imprecision in the commentary on the results which could lead a casual reader to become confused (see below)
- connections to existing results on differential privacy seems to be missing

additional comments:
- The restriction to local privacy, which is important for the results, makes the privacy model quite different than the differential privacy model, a fact which many readers may not appreciate. This confusion may be exacerbated by statements such as those at the bottom of page 4: "Thus, for suitably large sample sizes n, the effect of providing differential privacy at a level $\alpha$…" The authors should avoid making such overly broad (and perhaps incorrect) statements when describing their results.

- Is the restriction on alpha in Theorem 1 necessary? In particular, experimental results suggest that $\alpha \approx 1$ may be the most one can expect for certain learning problems (under differential privacy), so it is unclear the the bound tells us about this case.

- Some commentary on the possible choices of $\rho$ may be nice, so that readers can see how different utility measures can be captured by the analysis.

- How does this density estimator compare to the M-estimation procedure of Lei? This paper is not cited at all, but I imagine the authors should be aware of it.

- There are many other approaches to histogram estimation for discrete data. While randomized response achieves the optimal rate of convergence, how do these other algorithms stack up?

ADDENDUM AFTER REBUTTAL:
* I think the distinction between the population vs. sample statistics needs to be explained more clearly and more explicitly at the beginning of the paper (c.f. response to Rev.9)
* A comparison to related work (Lei and those brought up by another reviewer) is important for context.
* A closer inspection of [10], which has now appeared, makes me construe the additional contribution of this paper more narrowly. While the venues (and hence audiences) for this and [10] are different, the contribution of this paper is twofold: a careful exposition of the local privacy model, and bounds for density estimation. The latter are new but the former is essentially contained in [10].


Summary: This paper is a NIPS-formatted version of an ArXiV manuscript, and uses a LeCam-style argument to derive a lower bound on estimation algorithms that operate on private data when the algorithm is not trusted by the data holder. As a corollary, randomized response turns out to be an optimal strategy in some sense.

Submitted by Assigned_Reviewer_9

This paper studies minimax bounds for probability estimation under the constraint that the estimation must preserve privacy of the individual data. The authors consider a privacy definition called local privacy. They study two probability estimation problems, multinomial estimation and density estimation. In both problems, the authors show sharp minimax rates of convergence. They demonstrate that, for the discrete multinomial estimation problem, local privacy causes a reduction in the effective sample size quadratic in the privacy parameter alpha. Since alpha can often be seen as a small constant, the effective sample size is of the same order as the non-private case. For the density estimation problem, the authors demonstrate that the optimal rate for the non-private setting is no longer attainable if local privacy must be preserved.

Overall the results are very interesting. As far as I could read, the proofs are correct. To the best of my knowledge, the minimax bound for density estimation under privacy constraint has not been considered before.

My main comment is that the minimax bound of the multinomial estimation is closely related to previous works on the noise complexity for differential privacy, but there is a lack of mention. In particular, two papers consider highly relevant problems, Hardt&Talwar, On the geometry of differential privacy, STOC, 2010; and De, Lower bounds in differential privacy, TCC, 2012. These two papers study worst case lower bounds for the error of linear queries under the constraint of differential privacy. The probability estimation problem considered in this paper is actually a special case of the linear counting query studied in those two papers. Also, the measures used for the error are the same L_2 metric. The only difference is that this paper considers local differential privacy while Hardt&Talwar and De consider differential privacy. Local privacy posts stronger constraint than differential privacy, and therefore lower bounds for differential privacy are also lower bounds for local privacy. I am wondering if the bound for local privacy given in this paper improves over previous bounds for differential privacy.

For the density estimation problem, the statement of the result is a bit confusing. The authors state that the lower bound in the local privacy setting is higher than the non-private setting. But the minimax bound in this paper is for the special case that the density can be expanded with trigonometric basis. I am wondering if the lower bound for the non-private setting eq.(13) holds for the general Sobolev space as given in definition 1 or for the special case of trigonometric basis. It is a fair comparison only if the non-private lower bound holds for the trigonometric basis.

Additional comments to the rebuttal:

The feedback partially clarifies the relation to previous worst case lower bounds. But note the work of Nikolov, Talwar, and Zhang, The Geometry of Differential Privacy: The Sparse and Approximate Cases (STOC 2013) also considers Mean Square Error. I suggest the authors add their explanations and missing references to the paper.
Summary: This paper proves sharp minimax bounds for pmf and pdf estimation with privacy guarantee. The results are interesting and the paper is well written. But there is a lack of mention about known results on the noise complexity lower bound for differential privacy which are very relevant to this paper.

Submitted by Assigned_Reviewer_10

Local Privacy and Minimax Bounds: Sharp Rates for Probability Estimations
------------------------------------------------------------------------------------------------------
The paper deals with local privacy -- a setting in which each user outputs a \alpha-differentially private signal based on her own type and the signals of the other n-1 users, and reports the data curator that type. The paper analyzes this setting using the framework of min-max expected error -- the adversary picks a distribution over inputs (types for the n users) and we pick a \alpha-DP local scheme so to minimize the distance between an estimation derived from the original data and a similar estimation derived from the reported signals. (Typically, sums, aggregations or averages.) The authors then give a lower bound for the min-max rate for l_2 distance estimations and users drawn from a discreet distribution over d types. They show that the simple technique of randomized response achieves meets this bound, and therefore, it is optimal. The authors then proceed to analyze the problem of density estimations, where again the lower bound on the min-max rate is met by a perturbation scheme in which each person perturbs only her own type.

The paper is nice and important -- it shows that some classic procedure is the best we can attain, since it meets certain lower bounds. I think that the NIPS community would find it interesting, and I therefore recommend acceptance.

I do have a few reservations though. First, style-wise, the explanation regarding min-max bounds could have been simpler. The paper also has lots of cumbersome notations -- in particular, since all bound given apply for l_2 norm, couldn't the bounds be phrased w.r.t this norm? Secondly, I would have loved seeing a comment about l_1 norm, which is the more restrictive, or a comment relating the given bounds to l_1 norms, as well as other lower bound in differential privacy (which granted, apply more to a classical, non-local, setting). Lastly, it seems as though the min-max bound discussed are "tailored" for this local-privacy setting. I wish the authors could have considered a broader set of settings and give min-max lower bounds for them too. I do like however that whereas local-setting allows users to randomize the type they report based on the remaining n-1 users, it turns out that the simple scheme in which each person randomizes the report solely based on her type is optimal. I wish the authors would state that explicitly in the text.

****A new reservation: by now, the list of FOCS 2013 accepted papers has been published. And so, I now feel that the paper is now an extension of an existing paper, especially the results of Section 3. I therefore still recommend acceptance, but not as strongly as before.
Summary: The paper analyzes the framework of local privacy using the min-max rate bounds, give lower bounds on the rate and show that the upper bound is met by randomized response. A nice result.
Author Feedback

Author rebuttal: We thank the reviewers for their comments and insightful reading of
the paper. We will address all the reviewer's comments in the updated
manuscript; we respond to the major concerns below.

REVIEWER 10

The reviewer points out that our bounds are all L2-based; this makes
our more general minimax framework a bit of overkill. We agree, though
we contend that our setting immediately extends to other loss
functions. Indeed, an extension of our lower bound technique in
Theorem 1--which uses the same mutual information bound and packing,
so that the packing points are separated by \delta in L1-norm--gives
minimax lower bounds on L1 of

d / \sqrt{n \alpha^2},

while the standard minimax rate is \sqrt{d / n}. (This fact can be
proved either via Le Cam's method or Fano's inequality). The lower
bound is matched (again) by randomized response.

The reviewer's curiosity on bounds on the L1 mirrors our own, and we
will include a sketch of these results in the final version.

REVIEWER 7

The reviewer's comments on clarity are helpful, and we will definitely
clear these up in the final revision.

The restriction on \alpha in Theorems 1-2 is not completely necessary:
roughly, we can allow \alpha to grow as log(.5 + p(n) / \sqrt{2}),
where p(n) is an increasing polynomial in n with p(1) = 1. (Roughly
p(n) = n^{1/2} for the multinomial bound.)

The reviewer's comments on general \rho are similar to Reviewer 10's;
we will add some discussion here (for example, the L1 case).

The histogram estimator of Lei is similar to ours--it uses Laplace
noise--but is not locally private. Histogram estimators also cannot
achieve minimax convergence rates for smoother distributions, where
different estimators are necessary; in our case, these are the
orthogonal series estimators (often, one uses kernel density
estimators; it is not clear how to privatize those). Lei's
M-estimators have sub-optimal convergence rates (because they rely
first on a density estimation step).

Adding Laplace noise is also optimal for histogram estimation (and
hence density estimation for Lipschitz continuous densities); to
distinguish between other algorithms, randomized response, and Laplace
perturbations might require a more asymptotic analysis (to get sharp
constants) rather than our finite sample guarantees. That said, our
results are *optimal*, meaning that no procedure can get better than
(numerical) constant factor improvements.

We will add a deeper discussion of previous work in the updated
version--we are happy to hear of citations we may have missed. See
also our feedback to reviewer 9.

REVIEWER 9

In the revision, we will certainly add more discussion of previous
work as suggested, but we do believe that a significant issue is
being overlooked in the reviewer's comparison to previous work.
Hopefully, the following comments will help to situate our paper.

To the best of our understanding, much of the previous literature
on lower bounds under privacy is that researchers have bounded
errors from the *sample* estimator (or related sample-based quantities)
as opposed to the *population* quantity (or more generally, some parameter
at the population level). The relationship between these quantities
is precisely that addressed by the theory of statistical inference.
Indeed, it is worth noting that bounds on sample quantities versus
those on population quantities can be very different; such differences
drive much of the technical work in the literature on statistical inference.

For example, Hardt & Talwar and De both provide bounds on the quantity

sup_x E[||Ax - \hat{\theta}(x)||^2]

where x is the sample (i.e., the n observations, or the dataset) and
\hat{theta} is the differentially private estimator.

In contrast, we provide bounds on estimation of the *population*
quantity

E_\theta[||theta - \hat{theta}||^2]

and these bounds are not implied by any previous work that we know of
(including H&T and De). In contrast to this past work, our results
have parallels with classical results in statistical minimax theory
(see references [24,27,28] in the paper).

Otherwise, it is worth noting that our bounds are somewhat more
pessimistic than most bounds under differential privacy, as local
privacy is more stringent. For example, the lower bounds of H&T (when
translated into our notation) are that if theta(x) is the *sample*
estimator, then

sup_x E[||theta(x) - hat{theta}||^2] \ge d / (n^2 alpha^2).

Two points are important here: first, this result does not imply our
result (since it is a worst-case sample as opposed to distributional
result); and second, it allows for faster convergence rates than 1/n,
since the lower bound decreases as n^2.

As noted in our response to the other reviewers, our \rho allows more
general lower bounds than only L2. (We hope to add an example in the
revised version.)

Finally, to clarify some points regarding density estimation, the
non-private bounds hold not only with the trigonometric basis, but
with any orthonormal basis of L^2[0,1]. (For more details, see
reference [24], Theorem 2.9 and Corollary 2.4.) In any case, since our
lower bound holds for this restricted class--and non-private
estimators exist *achieving* the faster rate (13) for broader
classes--our lower bound shows a separation in rates regardless.